# Fiber Bragg Grating-Based Smart Garment for Monitoring Human Body Temperature

**DOI:** 10.3390/s22114252

**Published:** 2022-06-02

**Authors:** Xiujuan Wang, Yaming Jiang, Siyi Xu, Hao Liu, Xiaozhi Li

**Affiliations:** 1School of Textile Science and Engineering, Tiangong University, Tianjin 300387, China; xjzwang@outlook.com (X.W.); jiangyaming@tiangong.edu.cn (Y.J.); xsy18812733967@163.com (S.X.); 2R&D Department of Personal Protective Lifesaving Equipment, Aerospace Life-Supports Industries Ltd., Xiangyang 441003, China; 3Aviation Key Laboratory of Science and Technology on Life-Support Technology, Xiangyang 441003, China

**Keywords:** fiber Bragg grating (FBG) sensor, smart garment, sensor calibration, body temperature monitoring, optical fiber bending loss

## Abstract

Body temperature provides an insight into the physiological state of a person, and body temperature changes reflect much information about human health. In this study, a garment for monitoring human body temperature based on fiber Bragg grating (FBG) sensors is reported. The FBG sensor was encapsulated with a PMMA tube and calibrated in the thermostatic water bath. The results showed that FBG sensors had good vibration resistance, and the wavelength changed about 0–1 pm at a 0.5–80 Hz vibration frequency. The bending path of the optical fiber after integration with clothing is discussed. When the bending radius is equal to or greater than 20 mm, a lower bending loss can be achieved even under the bending and stretching of the human body. The FBG sensor, the optical fiber, and the garment were integrated together using hot melt glue by the electric iron and the hot press machine. Through experiments of monitoring human body temperature, the sensor can reach the human armpit temperature in about 10–15 min with the upper arm close to the torso. Because it is immune to electromagnetic interferences, the smart garment can be used in some special environments such as ultrasonography, magnetic resonance (MR), and aerospace.

## 1. Introduction

Body temperature is regarded as the first vital sign [1]. In daily life, it provides an insight into the physiological condition of a person. Either an elevated body temperature or a degraded body temperature indicate that the person is suffering from diseases. In the medical area, the continuous monitoring of body temperature can offer various kinds of information valuable for clinical diagnosis, and it is a useful guide to take suitable action [2]. Wearable flexible temperature sensors have been developed for monitoring body temperature [3,4]. They measure the temperature by the electrical signal changes of the thermo sensitive materials, which are caused by the temperature change. The electrical signal is easily affected by electromagnetics, so the sensors cannot be used in some special environments, such as ultrasonography, magnetic resonance (MR), and aerospace.

Fiber Bragg grating (FBG) sensors are sensitive to environmental variables, such as temperature, stress, bending, and pressure. They have been widely applied in the fields of civil engineering, the automotive industry, aerospace, the oil and gas industry, and biomedicine as high-precision sensors [5,6,7,8].

Compared with electrical and mechanical sensors, they are immune to electromagnetic interferences. These features make FBG sensors more of an emerging solution for the monitoring of physiological parameters, and they are particularly attractive for application in smart textiles. During the last decade, the garment, wearable straps, adhesive tape, or insoles that were embedded with in the FBG sensors were used to monitor body physiological parameters such as heart rate [9,10,11], body surface strain [12,13], blood pressure [14,15], respiration [9,10,16,17], Ballistocardiogram (BCG) signal [18], joint postures [19,20,21,22], plantar pressure [23,24,25,26,27], and temperature [28,29]. As reported in reviews [29], the garment was divided into several blocks, the FBG temperature sensor was woven into the block, and then the blocks were sewn together. The weaving process was complicated, and when the blocks were sewn together, the sensor may be damaged. In addition, the report did not discuss how to design the path of optical fiber, and this is important to transmit the signal.

In this article, an FBG-based smart garment was developed for monitoring body temperature. The principle, fabrication, and encapsulation of the FBG temperature sensor are presented in detail, and the sensor was calibrated and its performance tested. Then, a new method of integrating garment, sensor, and the path of optical fiber was designed, and the integrated garment was worn on the body for a temperature monitoring experiment.

## 2. Basics of Light Transmission in Optical Fibers and FBG Sensing

The advent of optical fiber with ultralow transmission loss has enabled optical communication to develop rapidly. It consists of a core, cladding, and coating, as shown in Figure 1a. While the core and cladding are responsible for light guiding, the coating protects the fiber from external influence. Basically, the refractive index of the core by doping with germanium is slightly higher than that of the cladding, and light transmission obeys the principle of total internal reflection at the interface of the core and cladding (Figure 1b).

Germanium dopant also introduces photosensitivity to the optical fiber core, which enables the Bragg gratings to be inscribed on it. The refractive index of the photosensitive fiber core is periodically modulated by the Bragg gratings [30,31]. When a broadband light is launched through the core of the optical fiber with Bragg gratings, the selective reflection of one particular narrow band of wavelengths, whose peak is termed as the Bragg wavelength or central wavelength, is observed, as shown in Figure 1c.

The Bragg wavelength is given by [32]
(1)$$\lambda_{B}=2n_{eff}\Lambda$$
where *λ_B_* is the Bragg wavelength reflected by the Bragg grating, *n_eff_* is the effective refractive index of the fiber, and Λ is the grating period. FBGs are sensitive to external perturbations such as strain and temperature. When the strain or temperature near the grating changes, the effective refractive index or/and grating period become changed, which leads to a shift in the reflected Bragg wavelength. The Bragg wavelength shift can be expressed by [32]:(2)$$\frac{\Delta \lambda_{B}}{\lambda_{B}}=\left(1-p_{eff}\right)\Delta \varepsilon +\left(\alpha +\beta \right)\Delta T$$
where *λ_B_* and Δ*λ_B_* are the initial Bragg wavelength and its change, and Δ*ε* and Δ*T* are the stain and temperature change. *p_eff_*, *α*, and *β* are the photo-elastic coefficient, thermal expansion coefficient, and thermal-optic coefficient of fiber, respectively. The property of the Bragg wavelength shift with respect to the external strain or temperature is exploited for its sensing applications [32,33].

In case the temperature of the external environment is constant, the Δ*T* = 0, and Equation (2) is as follows:(3)$$\frac{\Delta \lambda_{B}}{\lambda_{B}}=\left(1-p_{eff}\right)\Delta \varepsilon$$

In case the FBG is not affected by the strain, Δε = 0 and Equation (2) is as follows:(4)$$\frac{\Delta \lambda_{B}}{\lambda_{B}}=\left(\alpha +\beta \right)\Delta T$$

## 3. FBG Temperature Sensor Fabrication and Sensing Performance Test

### 3.1. Sensor Design and Encapsulation

The relationship between the change of the Bragg wavelength and the temperature at a constant strain is linear based on Equation (4). The temperature sensitivity of the bare FBG was about 10 pm/°C at about 1550 nm in the central wavelength [29], and the accuracy was about 0.1 °C, which basically can meet the needs of human body temperature measurement. In order to prevent any posture of the human body from causing strain sensing to the FBG when the sensor is worn on the human body, the sensor can be encapsulated in a hard object. At the same time, the sensor should not affect the comfort of the garment. Stainless steel, ceramic, and polymer materials are commonly used to encapsulate FBG. Stainless steel and ceramics have an excellent bending resistance and high temperature resistance, but stainless steel may cause discomfort when worn on the human body, and ceramics are brittle and easily damaged. Polymethyl methacrylate (PMMA), known as plexiglass, has the bending strength of 110 MPa and the thermal deformation temperature of 74–107 °C. Compared with stainless steel, it is safer and more comfortable when it is worn on the body. By comparing the shapes, the commercial PMMA tube with a 2 mm outer diameter, and a 1.1 mm inner diameter was chosen, and two structures were designed to encapsulate FBGs as shown in Figure 2. In Figure 2a, only one end of the optical fiber is adhered to the PMMA tube, and in Figure 2b, the two ends of the optical fiber are adhered to the ends of the PMMA tube, respectively.

In this study, a standard single mode optical fiber (Corning, SMF-28) was used for FBG fabrication. The fiber was loaded with hydrogen to render it photosensitive and striped off the acrylate coating of a small portion (about 10 mm), then an FBG of about 5 mm length was imprinted in the fiber core using the standard phase mask technique, and recoated with acrylic. The PMMA tube of 13 mm length was used as the single-ended encapsulated tube, and the one of 15 mm length was used as the duel-ended encapsulated tube. In order to make the size of the grating area smaller and maintain the stability and accuracy of the measurement, we designed the fiber Bragg grating measurement area to be 5 mm, the size of the fixed end of the FBG and PMMA was designed to be 5 mm to improve stability, and the only sealing end was designed to be 3 mm. For single-ended encapsulation, first the tail fiber was cut off, one end of the PMMA tube was sealed with glue, the optical fiber was inserted into the PMMA tube, and the other end of the PMMA tube and the optical fiber were glued together. For dual-ended encapsulation, the optical fiber was connected to the FBG interrogator and inserted into the PMMA tube, the grating part was placed in the center of the PMMA tube, the front part of the grating and the PMMA tube were sealed with glue, and the tail fiber was straightened with a clamp, the wavelength change in the FBG interrogator was changed, and when the change was about 0–1 pm, the optical fiber and the PMMA tube were glued together. More glue was added to the bonding end of the optical fiber and the PMMA tube to prevent the optical fiber from falling off. Through experiments, the length of the single-end encapsulation PMMA tube was set to 13 mm, as shown in Figure 2a, and the length of dual-ended encapsulation PMMA tube was 15 mm, as shown in Figure 2b. The optical fibers were covered with the ethylene-tetrafluoroethylene (ETFE) fluoropolymer as the protective jacket, which could make the optical fiber more flexible and stronger. The optical fiber, jacket, and the ends of the tubes were fixed with epoxy glue.

### 3.2. Calibration Experiment of Sensor

Sensors should be calibrated against a certified thermometer before use. Currently, there is no consensus on the best approach for FBG sensors’ calibration [34]. Some literatures have reported calibrations by immersing FBG sensors into a thermostatic water bath [35] or different temperature liquid (liquid nitrogen (−195.8 °C), ice slush (0 °C), and boiling water (100 °C)) with a platinum thermoresistance (Pt-100) element [36]. The calibration error of the former is related to the temperature accuracy of the water bath. The latter only calibrated three fixed points, which directly affected the calibration accuracy.

In this work, the block diagram of the experimental setup is depicted in Figure 3a. The thermostatic water bath was heated to different discrete temperatures. A platinum resistance thermometer (YET-720L, Kaipusen) with a reported accuracy of ±0.1 °C and a Pt-100 platinum resistance temperature sensor were used as the standard measure of the water bath temperature. The FBG sensor and the Pt-100 sensor were immersed in the same place of the water bath. At each discrete temperature, the values of the thermometer and FBG wavelength were recorded when they fluctuated about 30 s slightly around a fixed value. Three calibration experiments were carried out three times at different times.

Figure 3b,c shows the relationship between the wavelength and temperature of the single-ended FBG sensor and dual-ended FBG sensor, respectively. For the single-ended FBG sensor, it can be observed that the wavelength of the Bragg grating increased with temperature and the wavelength shifts of all three trial runs agreed well with each other, which shows that the sensors had a good stability and repeatability. However, for the dual-ended FBG sensor, the wavelength of the Bragg grating increased with temperature, but the wavelength shifts of all three experiments were not quite consistent. This may be caused by the change in the length of the PMMA tube (the coefficient of linear expansion is approximately (5~9) × 10^−5^/°C.) due to the change in temperature, which resulted in the strain applied to the FBG. Thus, further experiments were carried out with single-ended FBG sensors.

The results of the three calibrations were, respectively, fitted with linear equations. By combining the data from the three trials and using the mean for determining the calibration equation, it was expressed as Equation (5). The measured temperature response at a constant strain was nearly linear at about 10.1 pm/°C.
(5)$$T=9.8996778868\times 10^{-10}\times \lambda -1.53417.743538047\times 10^{-10}\;\;\;R^{2}=0.9999$$
where *T* is the temperature, and λ is the wavelength of the FBG sensor.

### 3.3. Anti-Vibration Performance Test of FBG Sensor

The human body is the source of vibration. The movement of the human body will cause the human body to vibrate, and even when the human body is at rest, the organs are also in vibration. The range of frequencies considered by the ISO for health, comfort, and perception is 0.5 Hz to 80 Hz [37].

When the FGB sensor is worn on the human body, the vibration of human body organs and movement will cause the vibration of the sensor. The article tested the impact of vibration on the sensor’s sensing performance. The test setup is shown in Figure 4a. The FBG sensor was fixed on the vibrator by adhesive tape, and the function waveform generator was used to adjust the vibration frequency. The linear power amplifier converts the signal input from the function waveform generator into a stable and adjustable excitation current, which is fed to the vibrator. Under the action of the excitation current, the vibrator generates an excitation force to make it vibrate. By setting a different vibration frequency from 0.5 Hz to 80 Hz, the wavelength change of the FBG sensor was recorded. The experiment was carried out in a constant temperature and humidity laboratory, in which the temperature change was about ±0.5 °C. The results in Figure 4b,c show that the wavelength of the sensor changed about 0–1 pm at any vibration frequency.

### 3.4. Temperature Calibration Verification

The result of the FBG sensor calibration was verified by two methods. One was the static temperature measurement; the other was the dynamic body temperature monitoring. The calibration formula (5) was input to the software by programming, which was used to convert the wavelength to temperature. At the room temperature about 22 °C, the water bath was heated to 25, 30, 35, 40, 45, 50, 55, and 60 °C. At every discrete temperature, the FBG sensor and the Pt-100 sensor were immersed in the same place. The temperature values of the pt100 and FBG sensor were recorded when they were stable. Figure 5a,b shows the values of the FBG sensor and Pt-100 sensor and the error of the FBG sensor relative to the Pt-100 sensor, and the absolute values of the errors are all less than 0.2 °C. Then, at the room temperature about 25 °C, the Pt-100 sensor and FBG sensor were placed in the armpit of the subject at the same time, and the sensors were clamped by the arm. After about 30 min, the sensors were removed from the armpit. The platinum resistance thermometer recorded one data per second, and the FBG interrogator recorded 10 data per second. By calculating the average value of the data recorded by the FBG interrogator per second, the temperature monitored by the two sensors is shown in Figure 5c, and the response time of the Pt-100 sensor and FBG sensor are 19 s and 3 s, as shown in insets of Figure 5c. Figure 5d shows the errors of the FBG sensor relative to the Pt-100 sensor when they were stable, and the absolute values of the errors are also less than 0.2 °C. The error of the FBG sensor mainly came from human body vibration, the calibration formula, and the accuracy of the platinum resistance thermometer.

## 4. Integration of Sensor and Garment

### 4.1. Integration Design

The mouth, axillary, and rectal thermometry are the common methods of human body temperature measurement in the medical field, of which the axillary thermometry is the most convenient. The skin surface of the human armpit is relatively flat, and the sensor can fully contact the human skin at this position.

In this work, the elastic tight-fitting inner vest was chosen as the FBG sensor carrier, which made the sensor easier and more comfortable to wear on the human body. There are many inner vests in different fabrics and styles on the market. A sleeveless inner vest, which was made of rib knitted fabric consisting of cotton and spandex, was chosen. The integrated solution of the inner vest and FBG sensor was designed based on the axillary thermometry, as shown in Figure 6. The FBG sensor was placed at the armhole’s lowest part of the inner vest (Figure 6a), which is also the closest part to the armpit when the vest is worn on the body. Optical fiber, which is used to transmit signals, is also important for the integration of the entire garment. If it is free, it may affect the signal transmission or be pulled off. Because the tight garment is worn on the human body, it may be stretched horizontally or vertically. When the optical fiber is integrated into the garment in a straight line, it is easily damaged as the garment is stretched. Therefore, the path of the optical fiber was designed as a curved line, as shown in Figure 6b. When the garment is stretched, the curvature of the optical fiber path changes, which can protect the optical fiber from being broken. The heat press machine was employed to glue the FBG sensor and optical fiber on the inner vest by the hot melt glue, as Figure 6c shows. Since the thermal deformation temperature of PMMA is about 68–69 °C, the FBG sensor was carefully glued to the vest with an electric iron in order to avoid damage to the PMMA tube.

### 4.2. Optimizing the Path of Optical Fiber

As Figure 7a shows, the bending of optical fiber modifies the guiding properties of the optical fiber and cannot meet the total reflection condition, resulting in an increase in its radiative (outpropagating) part and the loss of optical power, which affects the FBG sensing performance. Different math models [38,39,40,41] have been suggested to calculate the relationship between the bending loss and the radius of the curvature. The parameters such as refractive index, wavelength, and their variations under various experimental conditions (temperature) can affect the loss results, so almost every theoretical model is in disagreement with the real experimental results. However, all theoretical models have the same conclusion, that is, the bending loss increases as the radius decreases and as the number of wrapping turns increases within a certain bending radius. When the bending radius is smaller than the threshold value, the oscillation of the bending loss appears [41,42].

In this work, the bending loss of optical fibers was tested, and the schematic diagram of the set-up is shown in Figure 7b,e. The two ends of the optical fiber were, respectively, connected to the optical laser source (λ = 1550) and the optical power meter. For bending the fiber, a wooden cylindrical rod with different radii (R = 22.5, 20, 17.5, 15, 12.5, 10, 7.5, 6, 5, 4, 2.5 mm) was employed. The influence of the bending radius (R) on loss was investigated at first. The result in Figure 7c shows that the bending loss becomes bigger and increases exponentially when the bending radius (R) is less than 10 mm. Then the influence of the wrapping turn number (N) on loss was tested using the cylindrical rods with a radius R ≥ 10 mm, and up to 15 turns were investigated. As shown in Figure 7d, the bending radius R = 10 mm was the critical bending radius, and, with the bending radius R > 10 mm, the bending loss effect was small.

The path of the optical fiber was designed as the curved line in Figure 7e. Thus, the article also tested the bending loss of the path’s bending number (N). The path was printed on the paper, along which the optical fiber was gradually pasted. The result shown in Figure 7f is consistent with the loss of the wrapping turn.

The paper tested the AD value of the center wavelength after being bent five times. (1) Initial wavelength without bending. (2) When the bending radius was 7.5 mm, the central wavelength was the AD value. (3) The bending radius was 10 mm, and the center wavelength was the AD value. (4) When the bending radius was 12.5 mm, the central wavelength was the AD value.

The paper tested the AD (analog-to-digital) value of the FBG reflected wavelength after the fiber was bent five times. AD refers to the output value of the light intensity detected by the spectrophotometer through the analog-to-digital converter; the larger the value, the easier it is to detect the center wavelength. As shown in Figure 8, when the bending radius is 7.5 mm and 10 mm, the AD value fluctuates greatly. When the bending radius is 12.5 mm, the AD value wavelength is smaller.

From the above results, the bending radius (R) of the optical fiber must be bigger than 10 mm under various stretching conditions.

The bending radius became bigger with the garment was stretched in the vertical direction, and it became smaller with the garment stretched in the horizontal direction. Therefore, the change of the bending radius with the optical fiber stretched in the horizontal direction is very important to design its path. When the garment is worn, the movement of the body causes the garment to stretch horizontally. As Figure 9a shows, the maximum horizontal stretch rate of the garment caused by human breathing was 12–14%, and that caused by the bending of the torso was 16–18%. Four paths with different bending radii were designed. The fabric was manually stretched to different degrees, as Figure 9c shows. The stretched rate and the bending radius of the optical fiber were calculated by digital image processing technology. The bending radius decreased with the increase of the stretch rate, as shown in Figure 9b, and for an optical fiber with an initial bending radius R ≥ 20 mm, the bend radius was still greater than 10 mm after being stretched 25%. In fact, the optical fiber was pasted on the vest by the hot melt glue, which made the bonding part more difficult to stretch than the other part. Thus, when the appropriate size of the vest is worn on the body, the stretch rate of the optical fiber path will be smaller than that of the manual stretching. From these results, the bending radius of the optical fiber path was designed to be equal or greater than 20 mm.

### 4.3. Body Temperature Monitoring Test

According to the above conclusions, the FBG sensor and the optical fiber with a bending radius of 20 mm was fixed to the vest with a needle and thread, and then they were integrated together using the electric iron and the hot press machine, as shown in Figure 10a.

At the room temperature of about 23 °C, the subject put on the inner vest, a long-sleeved T-shirt, and a coat, in order, and the sensor was connected with the FBG interrogator. Then, the Pt-100 sensor was stuck in the armpit using adhesive tape to keep it from slipping. The arm was close to the torso to reduce underarm air exchange.

The temperature monitoring curves of the FBG sensor and the Pt-100 sensor are shown in Figure 10b. The Pt-100 sensor was placed under the human armpit and began to change steadily after about 25 s. The FBG sensors were affected by the microclimate inside the garment and began to change steadily after about 7–10 min. After about 10–15 min, the temperature values of the two sensors were essentially coincident. Figure 10c shows the error of the FBG sensor relative to the Pt-100 sensor, and the absolute values were less than 0.2 °C. The error was mainly caused by two aspects. First, the accuracy of the platinum resistance thermometer was ±0.1 °C, which led to errors in the calibration process of the FBG sensor. The other is that, when wearing clothes to monitor the body temperature, the vibration of the human body may cause errors in the FBG sensor, which was about 0–1 pm (0–0.1 °C).

Figure 10d, e shows that the swinging arm did not affect the monitored human body temperature. Therefore, when wearing the vest to monitor body temperature, the arm can perform some movement.

## 5. Conclusions

This paper demonstrates the principle and application of FGB sensing technology to design and develop an inner vest for monitoring human body temperature. The FBG sensor was ergonomically designed. It was encapsulated and calibrated, and it tested the antivibration performance in the laboratory. Its response time and resolution were 3 s and 10.1 pm/°C, respectively. It was placed at the armhole’s lowest part of the inner vest. The path of the optical fiber was designed in great detail, and the bending radius should be equal to or greater than 20 mm. Through the human body temperature monitoring experiment, the error between the FBG sensor and the calibration sensor was ±0.2 °C.

The inner vest integrated with the FBG sensor is very convenient for monitoring hu-man body temperature. Because the FBG sensor is waterproof because of its encapsulation, it can be washed with the right washing method. Compared to electronic sensors, FBG sensors are immune to electromagnetic interference, so the inner vest can be used to monitor the patient’s temperature during an MR or ultrasound, as well as the pilot’s temperature. As the modern electromagnetic environment is becoming complex, its application will become more and more extensive.

## Figures and Tables

**Figure 1 sensors-22-04252-f001:**
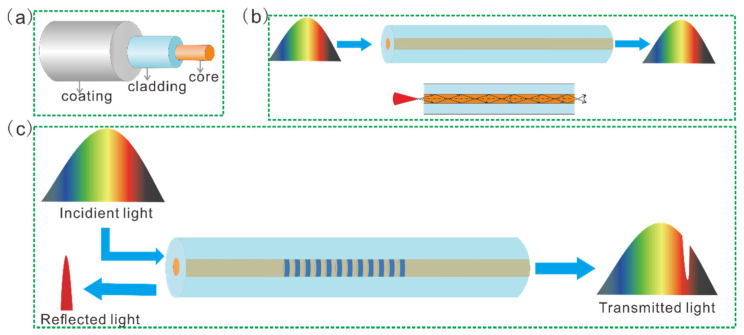
Working principle of the fiber Bragg grating (FBG). (**a**) optical fiber structure, (**b**) principle of total internal reflection, and (**c**) principle of FBG light Transmission.

**Figure 2 sensors-22-04252-f002:**
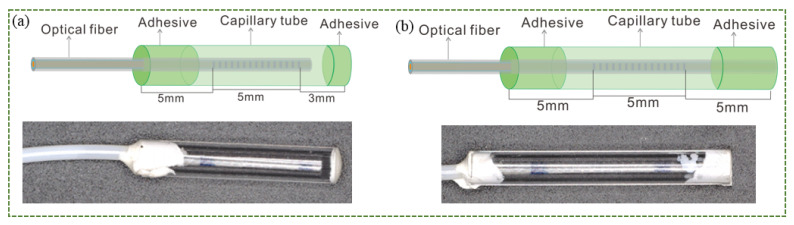
The structure of the sensors, (**a**) single-ended encapsulation, and (**b**) dual-ended encapsulation.

**Figure 3 sensors-22-04252-f003:**
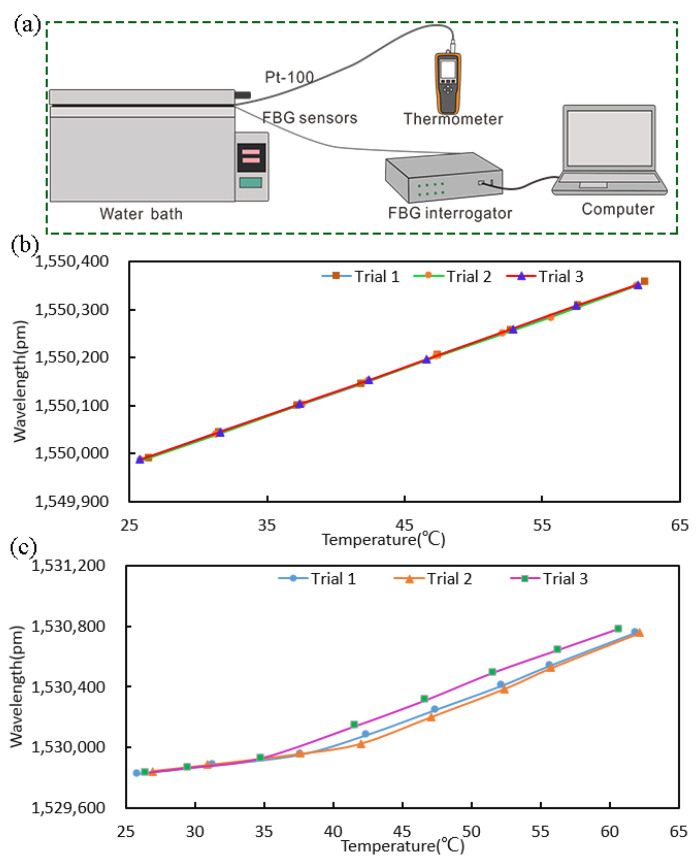
The calibration experiments. (**a**) The calibration experiment setup. (**b**,**c**) The relationship between wavelength and temperature of the single-ended FBG sensor and dual-ended FBG sensor, respectively.

**Figure 4 sensors-22-04252-f004:**
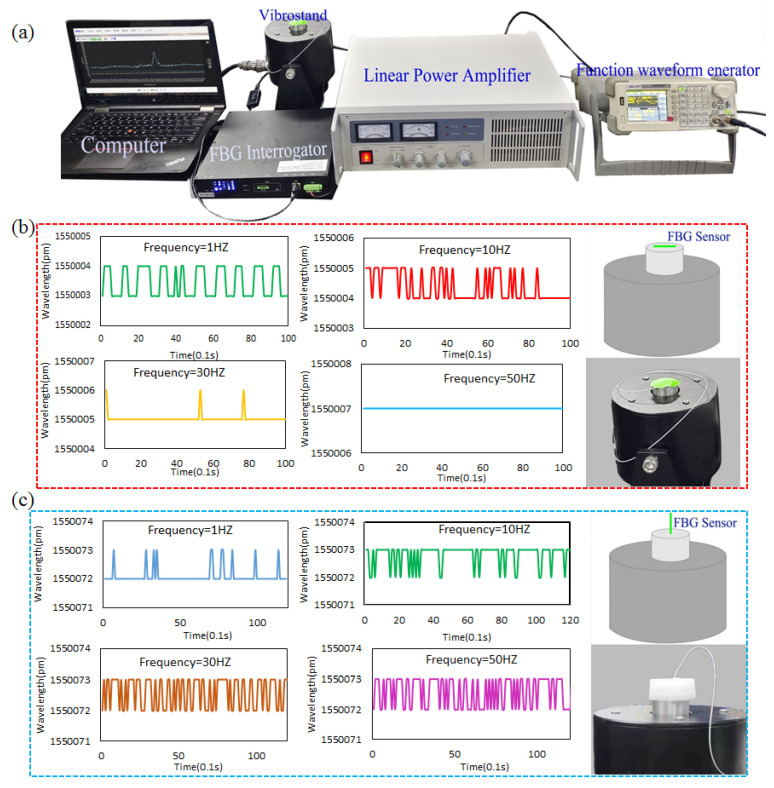
Anti-vibration performance test of FBG sensor, (**a**) the experiment setup, (**b**,**c**) the wavelength changes with different vibration frequencies.

**Figure 5 sensors-22-04252-f005:**
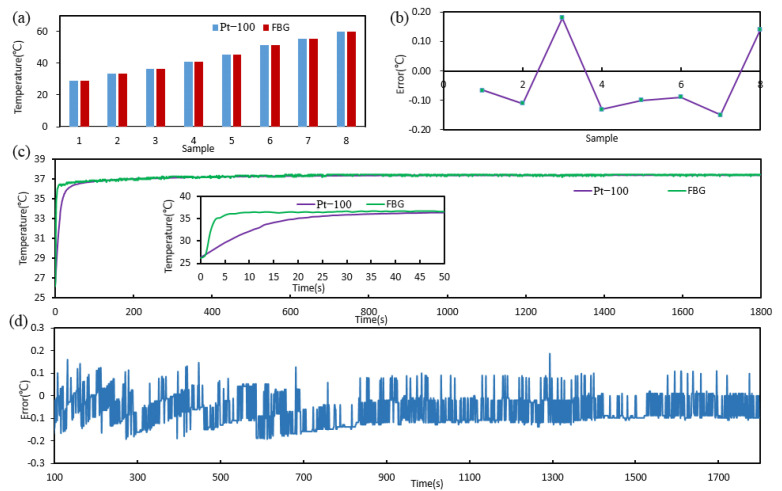
Temperature calibration verification and its error of the FBG sensor. (**a**) The static temperature measurement by the FBG sensor and the Pt-100 sensor, and (**b**) the errors of the FBG sensor relative to the Pt-100 sensor. (**c**) The body temperature monitoring of about 30 min by the FBG sensor and Pt-100 sensor, and (**d**) the errors of the FBG sensor relative to the Pt-100 sensor.

**Figure 6 sensors-22-04252-f006:**
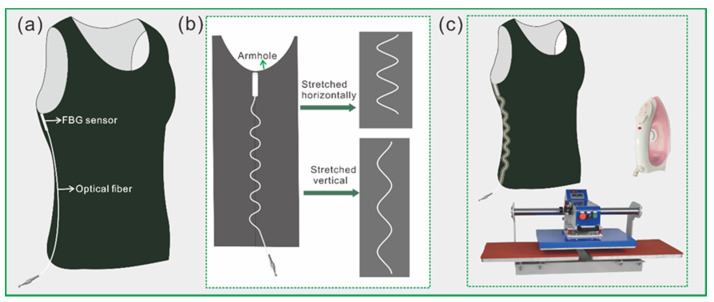
The integrated solution of the garment and FBG sensor. (**a**) The position of the FBG, (**b**) the path design of optical fiber, and (**c**) the method of the vest, the FBG sensor, and the optical fiber integration.

**Figure 7 sensors-22-04252-f007:**
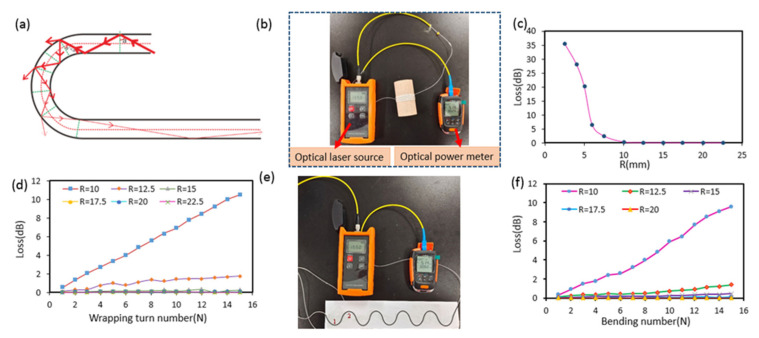
The bending loss test of the optical fiber. (**a**) The light guiding properties when the optical fiber is bent. (**b**) Schematic diagram of the optical set-up for measuring the loss of optical fibers with different radiuses and wrapping turn numbers. (**c**,**d**) The variation of loss against the bending radius and wrapping turn number. (**e**) Schematic diagram of the optical set-up for measuring the loss of optical fibers with different curves. (**f**) Variation of loss against the path bending number.

**Figure 8 sensors-22-04252-f008:**
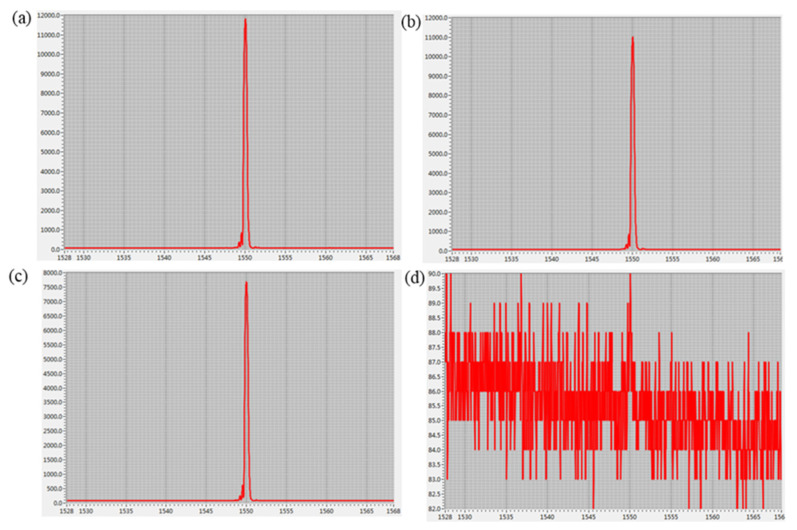
The relationship between fiber bending and the AD value fluctuate of the FBG reflected wavelength. (**a**) The optical fiber is freely extended, (**b**) the bending radius is 12.5 mm, (**c**) the bending radius is 10 mm, and (**d**) the bending radius is 7.5 mm.

**Figure 9 sensors-22-04252-f009:**
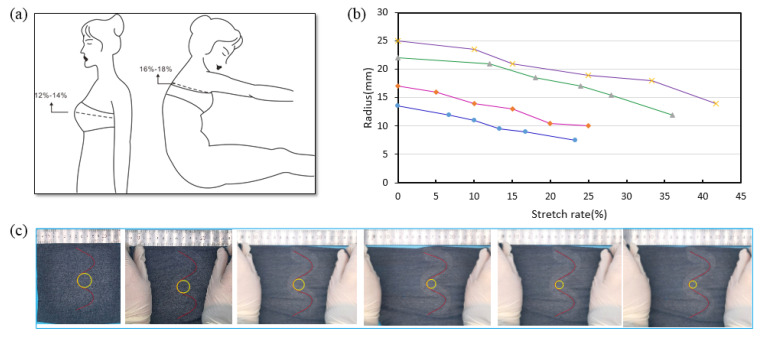
The relationship between the horizontal stretching of the optical fiber path and the bending radius. (**a**) The effect of human movement on the maximum horizontal stretch rate of the garment. (**b**) The relationship between the bending radius and the stretch rate with a different initial bending radius. (**c**) The manual stretching test of the optical fiber path.

**Figure 10 sensors-22-04252-f010:**
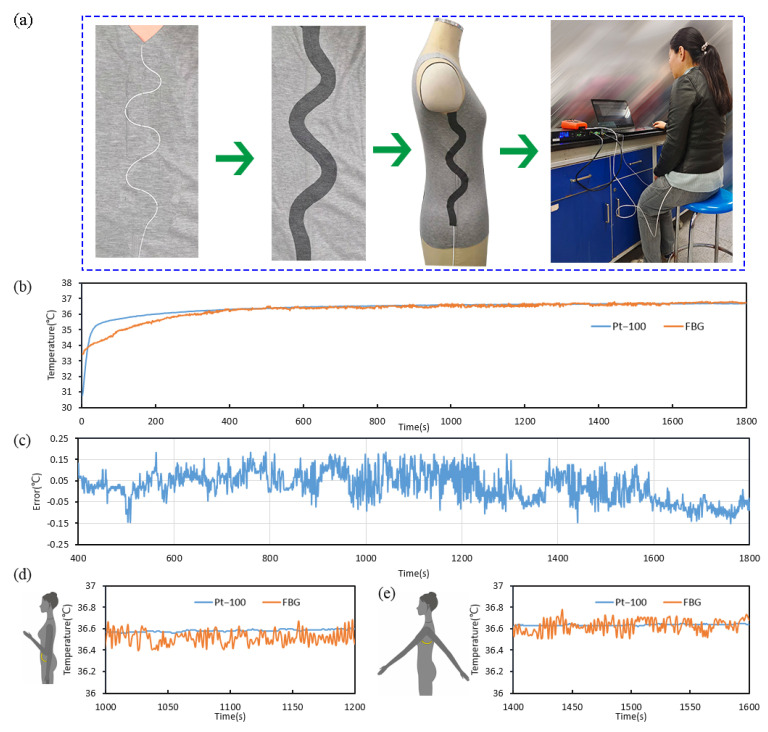
Body temperature monitoring test by being worn on the inner vest. (**a**) The integration process of the inner vest. (**b**) The body temperature monitoring curves of the whole process. (**c**) The error of the FBG sensor relative to the Pt-100 sensor. (**d**,**e**) The human body temperature when the arm swings.

## Data Availability

Not applicable.

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
