# Peer review of "Fiber Bragg Grating-Based Smart Garment for Monitoring Human Body Temperature"

_sensors, 2022, doi:10.3390/s22114252_

Round 1
Reviewer 1 Report
The paper presents a solution for monitoring body temperature using fiber Bragg gratings. This approach does not have a clear novelty, but in their work, the authors approached the study of this issue in a comprehensive manner. The following should be noted as remarks.
1. The authors should explain what caused the resulting temperature measurement error (Figure 10c-d).
2. What limit value can be obtained, and what is it limited to?
3. It is not clear from the text of the work which version of the casing was used in the final sample (pasted on both or one side into a PMMA tube). If the insert was on both sides, then it would be worthwhile to investigate the repeatability of the calibration characteristics after several heating / cooling cycles.
Author Response
Many thanks for your comments concerning our manuscript entitled “Fiber Bragg grating-based Smart Garment for Monitoring Hu-man Body Temperature” (ID: sensors-1714699). They are valuable and very helpful for revising and improving our paper. Now we have carefully revised the manuscript according to your comments, as explained below.

Reviewer 2 Report
Overview: The authors present an interesting account of design and characterization of an FBG-based temperature sensor embedded in a garment. The reviewer considers this work interesting, relevant to field, and pushing the envelope of current technologies. This work can be considered for publication after the authors will address the queries listed below.
Major queries:
- In lines 116-120 when describing the sensor, it is necessary to provide more information about single-ended and dual-ended encapsulation. What is the rational behind these configurations?
- In lines 155-158: “The results of the three calibrations are respectively fitted with linear equations, and 155 the calibration formula with the largest correlation coefficient is selected and it is ex-156 pressed as equation (5). The measured temperature response at a constant strain is nearly 157 linear about 10.1 pm/â—¦C.”
The reviewer believes that proper practice is to combine the data from 3 trials and use the mean for determining the calibration curve, instead of selecting the one curve out of three based on largest R-value.
- Regarding the vibration experiment in lines 161-179: this experiment appears to consider only a single vibration direction, which is transverse to the axis of the fiber. The reviewer believes that it makes sense to perform an additional experiment where the sensor will be oriented such that the axis of the fiber will be aligned with the direction of motion.
- From Figure 10(b) it is evident that the FBG-based sensor requires longer time to reach steady state readings. The paper will benefit greatly if the authors will add some discussion on how that aspect of sensor performance can be improved.
- How does perspiration affect temperature readings of the sensor?
- Please mention the sampling rate for the body temperature test described in Figure 10. Did the authors use any digital filtering to remove high frequency components?
- The reviewer believes that “Conclusions” section shall be improved by including relevant quantitative characteristics and results from experimental tests, as well as
Minor queries:
- Figures 3b and 3c – “trail” shall probably be “trial”.
- Define “Y” and “X” in equation (5). It is necessary to use previously defined variables, e.g. wavelength was defined as “lambda”.
- Time axis is difficult to see and read in Figure 5(d) and 10(c). Please revise it.
- Please consider using an English editing service to correct certain sentences and word choices to improve readability of the paper.
Author Response

(The authors gave the same response as above.)

Round 2
Reviewer 2 Report
Thank you for addressing reviewer's suggestions to improve the manuscript.